# Research on local sound field intensity control technique in metasurface based on deep neural networks

**Huanlong Zhao, Qiang Lv** **\*, Zhen Huang, Wei Chen, Guoqiang Hao**

School of Electrical and Electronic Engineering, Wuhan Polytechnic University, Wuhan, 430048, China

\* whpu_Lv@hotmail.com

## Abstract

The use of tunable metasurface technology to realize the underwater tracking function of submarines, which is one of the hotspots and difficulties in submarine design. The structure-to-sound-field metasurface design approach is a highly iterative process based on trial and error. The process is cumbersome and inefficient. Therefore, an inverse design method was proposed based on parallel deep neural networks. The method took the global and local target sound field feature information as input and the metasurface physical structure parameters as output. The deep neural network was trained using a kernel loss function based on a radial basis kernel function, which established an inverse mapping relationship between the desired sound field to the metasurface physical structure parameters. Finally, the sound field intensity modulation at a localized target range was achieved. The results indicated that within the regulated target range, this method achieved an average prediction error of less than 5 dB for 92.9% of the sample data.

**Data Availability Statement:** We have uploaded the minimal anonymized data set necessary to replicate our study findings to a stable, public repository. This is the link where one can access

## 1. Introduction

Acoustic super-surface materials, utilizing the technology of modifying material properties based on the generalized Snell's law, possess exceptional acoustic control abilities at subwavelength scales. Therefore, they have emerged as research hotspots in areas such as acoustic lenses, noise reduction, and acoustic stealth [1–4]. Pentamode acoustic metasurface can further compensate for the narrow-band limitations of traditional metasurfaces [5,6]. So far, researchers have achieved some sound field modulation works through the use of various structural designs such as Helmholtz resonators, coiled channels, mazes, and cavities [7–13]. Also, the researchers have utilized combinations of different materials, such as a combination of water and silicone rubber or polyurethane composites, to achieve the goal of reflective sound field modulation [14,15], other researchers use bottom-up inversion optimization algorithms to design metasurfaces [16–19]. While these phase mutation methods can achieve acoustic field modulation [20–25], it is noteworthy that all of the aforementioned methods have been studied based on the forward model of the sound field and finite element simulation. This research process requires a substantial amount of time from the researchers.

that information: https://github.com/bjhkbj/dataset.
git.

**Funding:** The National Natural Science Foundation of China (Grant No. 61873101), the PetroChina Innovation Foundation (Grant No. 2020 D-5007-0305), and the Marine Defense Technology Innovation Center Innovation Fund (Grant No. JJ-2020-719-03-02) supported this study.

**Competing interests:** The authors declare no conflict of interest.

Considering the limitations of conventional metasurface research methods, machine learning methods have been applied to the acoustic metasurface inverse design process, which enabled the optimization of structural parameters. Some results have been achieved [26–28]. Among them, Zhao et al. used a convolutional neural network model to establish a mapping of the local acoustic field to the phase gradient of the metasurface to achieve regional control of the local acoustic field [27]. Li et al. proposed a tandem neural network approach to reverse-engineer the phase of a metasurface unit such that the energy loss of an acoustic wave in the return direction is greater than 10 dB [28]. Long, Chen et al. used genetic algorithms to respectively design metasurface structures for sound absorption [26,29]. Li, Lin, et al. have respectively used machine learning for encoding metasurfaces to enable the modulation of the sound field by arranging these logical units into specific sequences [30,31]. These studies have taken advantage of the benefits of machine learning methods for model construction, which can help weaken complex physical mechanisms and reduce the need for model accuracy. This shows that the addition of deep learning is relevant to metasurface modulation techniques.

This paper introduces a novel metasurface inverse design method leveraging parallel deep neural networks (PDNN). The method respectively extracts the key information of the acoustic field and the metasurface as the input and output of the PDNN network. With the help of the kernel loss function and the constraint performance provided by the constraint network, it establishes an inverse mapping relationship between the target acoustic field and the physical structure parameters of the super-surface. Model validation show that this method can realize the regulation of local sound field intensity. This may be a novel way to achieve stealth for submarine vehicles.

## 2. Physical model of the metasurface local sound field

Turing the process of realizing intensity modulation of the target acoustic field, a physical model of the metasurface local sound field was used to acquire the dataset. The pentamode metasurface was chosen for sound field simulation because of its advantages of impedance matching with the ambient medium and wide frequency [32]. For a pentamode metasurface based on the generalized Snell's law, the material density distribution $\rho(x)$ is the decisive parameter affecting the reflected acoustic field. When the sound wave is vertically incident on the pentamode metasurface with the acoustic velocity $c_0$, its ideal density distribution $\rho(x)$ satisfies Eq (1) [33]:

$$\rho(x) = (\sin(\theta_r)x/2h + C_0)\rho_0, \quad 0 \leq x \leq L \tag{1}$$

where $L$ is the length of the metasurface, $C_0$ is the integration constant, $\theta_r$ is the reflection angle, $\rho_0$ is the density of incident medium, $h$ is the normal thickness of metasurface, and $x$ is the position. Artificial periodic structures cannot realize a continuous material density distribution on the theoretical metasurface. To approximate this continuous distribution, the theoretical metasurface can be discretized into n cells along the length ($i = 1, 2, \ldots, n$). The density of each discrete cell is characterized by the density $\rho_i$ at its center position [34,35].

In this paper, we started from the idea of parametric modeling without structural constraints on the metasurface structural units. Simplified parameters were used instead of metasurface structural units. With the method of unit combination, the phase mutation was adjusted at the same time to realize the local tuning of the acoustic field. The hypersurface has a normal thickness of 0.12m and a length of 1m. In the example of a sonar-detecting submarine shown in Fig 1(A), the incident acoustic wave can be viewed as a plane wave. When the incident acoustic wave contacts the surface of the submersible, the main reflected acoustic field is adjusted from the 90° direction to the other direction through the acoustic field

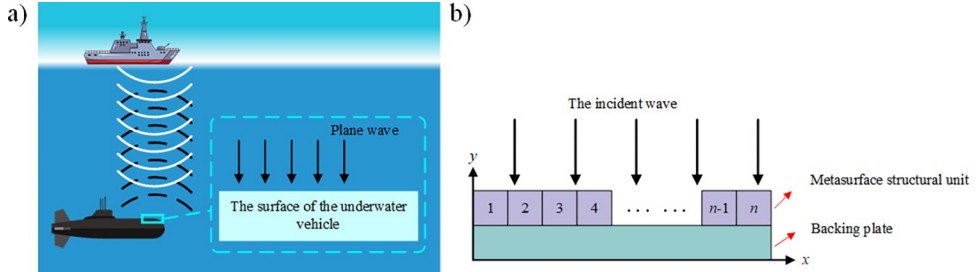

**Fig 1. a)** Schematic of the sound field model. **b)** Schematic of the pentamode metasurface.

modulation technique. The intensity of the acoustic field in the return direction is also changed. Therefore, we established the physical model of the sound field following the approach illustrated in Fig 1(B). A plane wave incident vertically underwater is used as the background field. The metasurface covered the backing plate surface and consisted of $n$ meta-surface structural units arranged along the x-positive direction. When a plane wave is incident on the metasurface in the y-reverse direction, the reflected waves generated by the $n$ metasur-face structural units interacting with each other make up the entire reflected acoustic field. The physical structural parameters of each structural unit could be different. In this paper, the physical structural parameters of each unit were obtained according to a gradient arrange-ment, which satisfied the requirements of the intensity characteristics of the desired sound field distribution, set $n$ to 25, so the length of each hypersurface structural unit is 0.04m.

## 3. Inverse design method based on parallel deep neural networks

### 3.1. Extraction of sound field features

In order to predicted the intensity of the target sound field, this paper drew on the idea of multi-scale that was to extract the global and local feature information of the sound field. The goal is the modulation of the local acoustic field, but the entire reflected acoustic field is a joint action of all metasurface structural units. In particular, the coupling relationship between indi-vidual structural units will have a significant impact on the reflected acoustic field characteris-tics. This coupling relationship will greatly increase the complexity of the model [36]. Therefore, when using local sound field intensity values as parallel deep neural network inputs, it was necessary to include feature information of the global sound field to constrain this pre-diction process.

The prediction of target sound field strength required the selection of global and local fea-ture information. According to Fig 2, it can be seen that the main change features of the reflected sound field are concentrated in the vicinity of the main reflection angle, the wave crest and trough, so the selection of global feature information can be extracted at the main change features. The local sound field intensity information was the value of the sound field intensity within the selected tuning target. After the global and local sound field feature infor-mation was extracted, it was used as an input to the parallel deep neural network.

### 3.2. Network model building

In this paper, a parallel deep neural network based on a fully connected architecture was used to predict the physical structural parameters of metasurface structural units. The model inputs were the extracted global and local feature information. The outputs were the density $\rho$ and the gradient value $g$ of the first structural unit. The density distribution of the entire

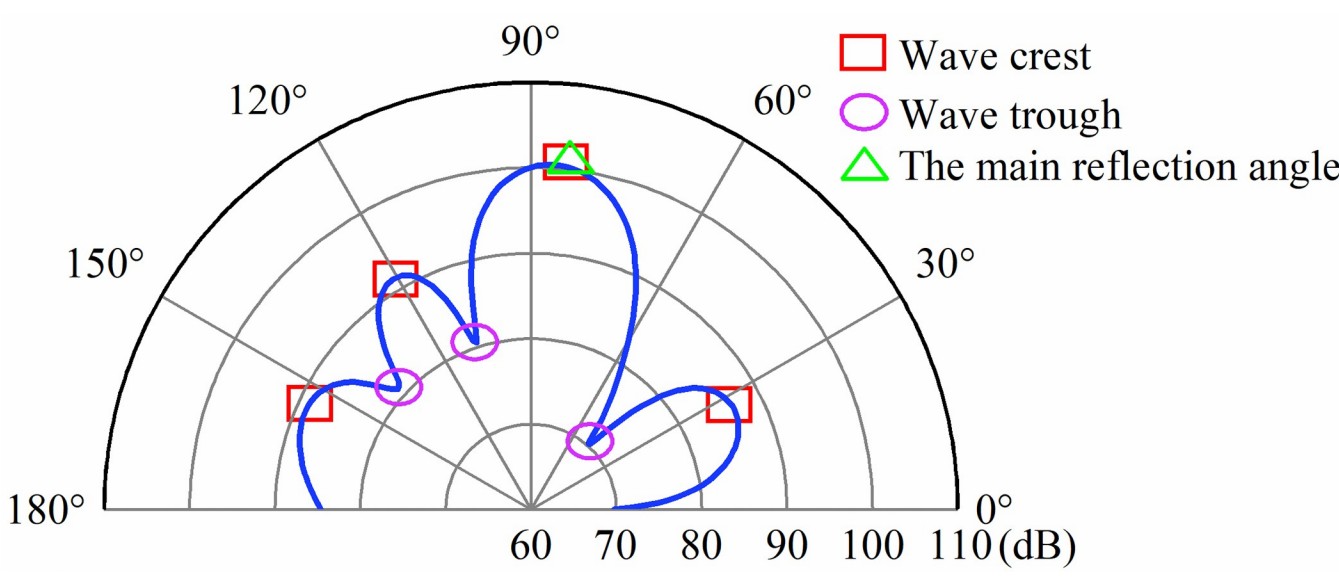

**Fig 2. Intensity distribution of the sound field.**

metasurface structural unit could be determined from the density $\rho$ of the first structural unit and the gradient value $g$ [32]. As shown in Fig 3, the network model consists of two sub-networks, which are a constraint network with global sound field feature information as input and a prediction network with local sound field intensity information as input. The parameter

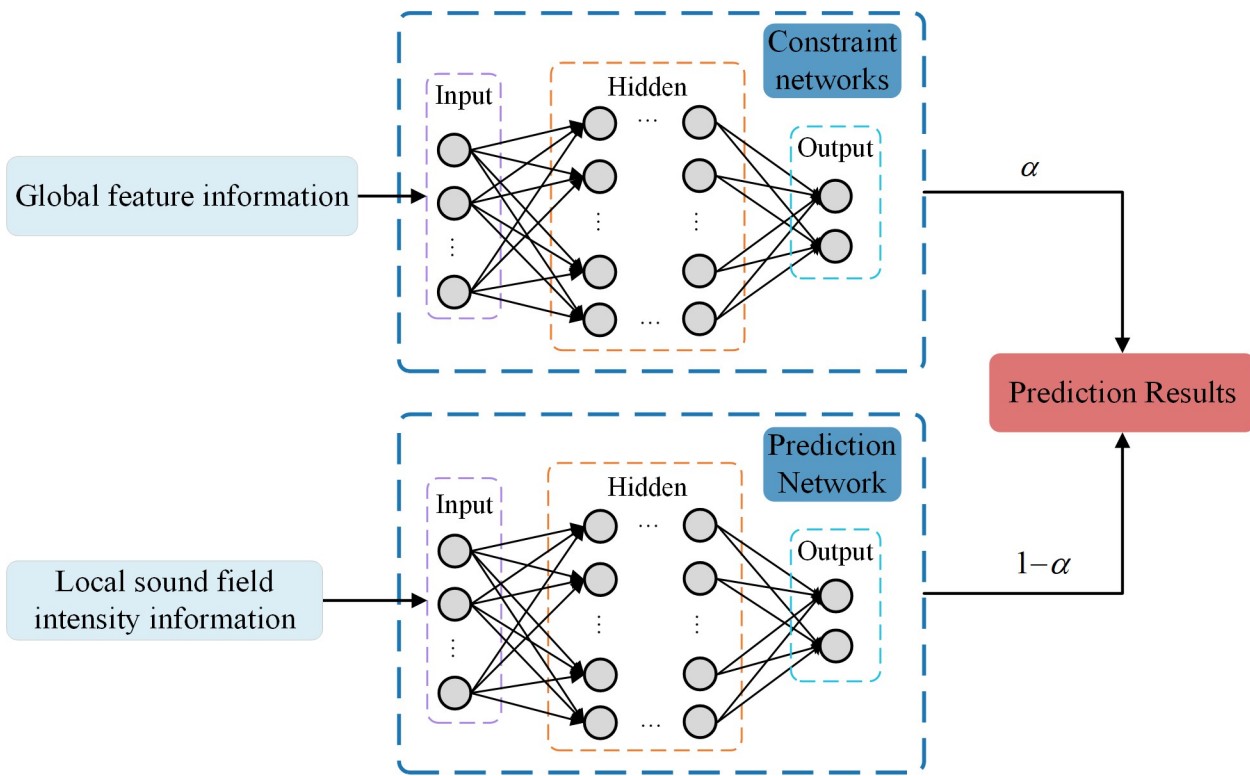

**Fig 3. Structure of the parallel network model.**

$\alpha$ in the figure was the weight factor, which was used to determine the weight of the constraint network in the overall model and took values in the range of [0,1]. The specific neural network topology consisted of an input layer, a hidden layer, and an output layer.

In the loss function selection, the MSE function is usually adopted as the loss function. However, the MSE loss function cannot accurately assess the nonlinear characteristics of the error and is sensitive to outliers. This problem can be solved by modifying the loss function using the radial basis kernel function. The modified loss function $L_{Kernel\text{-}MSE\text{-}Single}$ can be written as Eq (2) [37]:

$$L_{Kernel-MSE-Single} = \frac{1}{N}\sum_{t=1}^{N}[1 - \exp(-(y_t - \hat{y}_t)^2/2\sigma^2)] \tag{2}$$

where $N$ was the number of samples and $\sigma$ was the parameter of the loss function itself, $y_t$ and $\hat{y}_t$ respectively represented true and predicted values. We set $\varphi = (y\text{-}\hat{y})^2$ and $\lambda = 2\sigma^2$, where $y$ was the true value and $\hat{y}$ was the predicted value, so $\varphi$ was the squared error between the true value and the predicted value. According to Eq (2) and Fig 3, the final loss function $L_{K\text{-}MSE}$ expression was shown in Eq (3):

$$L_{K-MSE} = \frac{1}{N}\sum_{t=1}^{N}\{\alpha[1 - \exp(-\varphi_{t1}/\lambda_1)] + (1 - \alpha)[1 - \exp(-\varphi_{t2}/\lambda_2)]\} \tag{3}$$

where $\varphi_{t1}$ and $\varphi_{t2}$ respectively represented the squared errors of the true and predicted values of the constraint network, the prediction network. $\lambda_1$ and $\lambda_2$ were respectively the number of input features for the constraint and prediction networks. The modified loss function computed the gradient to the network parameters and completed the update to the network parameters.

## 4. Analysis of model validation results

### 4.1 Dataset preparation and setup parameters

The specific composition process of the dataset was shown in Fig 4, which consisted of two parts: labels and inputs. The labeling part consisted of the first metasurface unit density $\rho$ and gradient value $g$. We randomly generated 30,000 sets of first block metasurface structural unit densities $\rho$ and gradient values $g$. The metasurface density distributions could be calculated from the generated data, which were inputted into a physical model of the acoustic field to derive the corresponding acoustic field intensity distributions. The input parts were global sound field feature information and local sound field intensity information, which could be obtained by feature extraction of the sound field intensity distribution. In Fig 1(A), the reflected wave will be reflected along the echo direction ($y$ reverse direction) when there is no metasurface, so the global sound field range could be set to 0˚~180˚. The energy of the reflected wave is mainly concentrated in the direction of the echo [38]. Therefore, we set the target regulation interval as 85˚~95˚. When extracting the sound field intensity values within the tuning target range, we sampled at 0.5˚ equal intervals with a dimension of 1×21. The sound field in the range of 0˚~180˚ was divided into 6 intervals at equal step. The rules for extracting global sound field features were as follows:

Step 1, Within each interval, select: 1. The sound field intensity values and angles of the maximum crest and two adjacent points on each side are required, 2. The sound field intensity values and angles of the minimum trough trough and two adjacent points on each side are required.

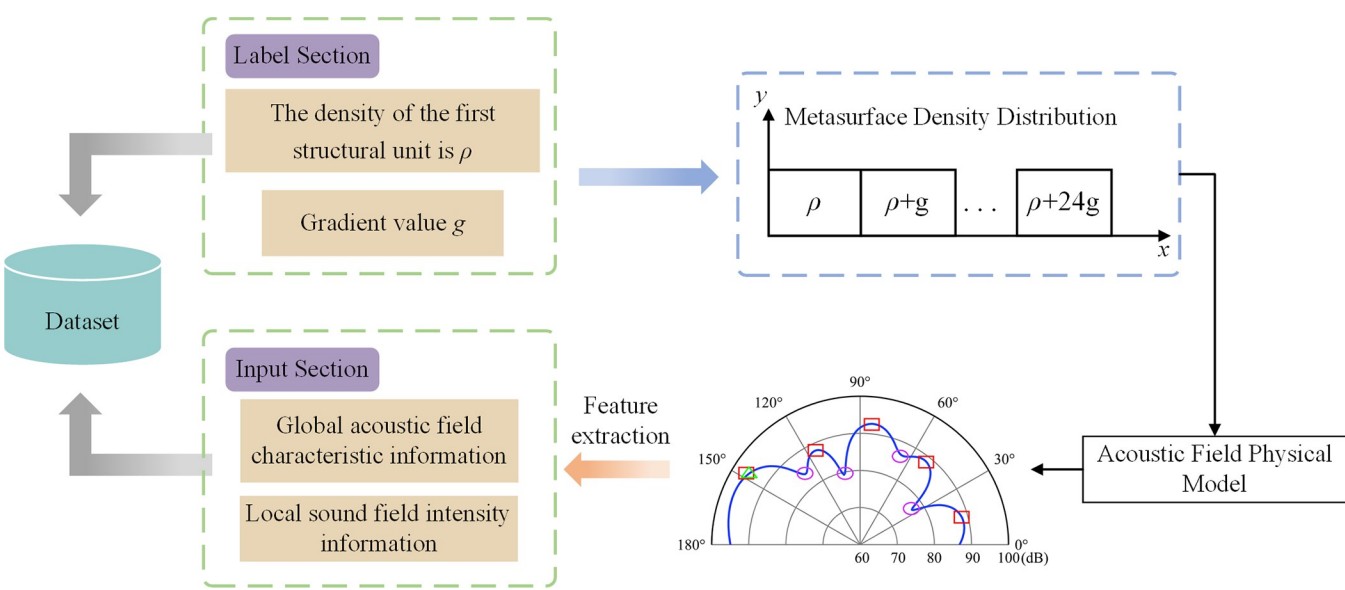

**Fig 4. The composing process of the dataset.**

Step 2: Within the global acoustic field, the act of selection: 1. The intensity value of the main reflection angle sound field, 2. The number of maximum peaks.

The above features were selected to characterize the global feature information of the sound field with a dimension of 1×122.

The number of neurons for the neural network model was set as shown in **Table 1**. The relevant parameters for training the network model were set as shown in **Table 2**.

## 4.2 Network model training

In network model training, the constraint performance of the constraint network directly affects the final network prediction results. If the constraint network weight α was too large means that the network model prediction results were more biased towards the global sound field distribution. Thus, the target sound field intensity prediction was not accurate enough. If the constraint network weight $\alpha$ was small means that the link between the target sound field intensity and the global sound field was weakened, and the coupling relationship between multiple structural units couldn't be learned during the model training process. The prediction results will also be inaccurate. Therefore, a key point in realizing local sound field intensity prediction was to determine the optimal weighting of the constraint network in the overall neural network by adjusting the weighting factors.

In this paper, the value of weight factor $\alpha$ was discussed. Under the unchanged conditions of the aforementioned parameter settings, model training and generalization ability verification were performed with different values of $\alpha$ (0.1, 0.2, 0.3, 0.4, 0.5, 0.6, 0.7, 0.8, and 0.9). The

**Table 1. The establishment of neuronal population size.**

| Neuronal layer | Predictive network | Constrained network |
|---|---|---|
| Input layer | 21 | 122 |
| Hidden layer | 100、300、600、1000、500、80 | 724、1000、543、200、100 |
| Output layer | 2 | |

Table 2. Parameter configuration of predictive models.

| Classification | Parameter name | Parameter Settings |
|---|---|---|
| Dataset partitioning | Training set | 24000 |
| | Testing set | 6000 |
| Network Configuration | Learning rate | 0.005 |
| | Activation function | LeakyReLu |
| | Optimizer | Adam |
| | Dropout | 0.1 |
| Training Process | Epoch | 800 |
| | Batch size | 64 |

variation of the loss function is shown in **Fig 5**. When the model convergence at different values of $\alpha$ was stabilized, the last 20 batches after the convergence of the network model were taken as the horizontal coordinates. **Fig 5** shows that the loss function value is decreasing as the value of $\alpha$ increases.

The loss function value can reflect the network model performance to some extent. Generalization ability is also one of the important indexes to evaluate the performance of network

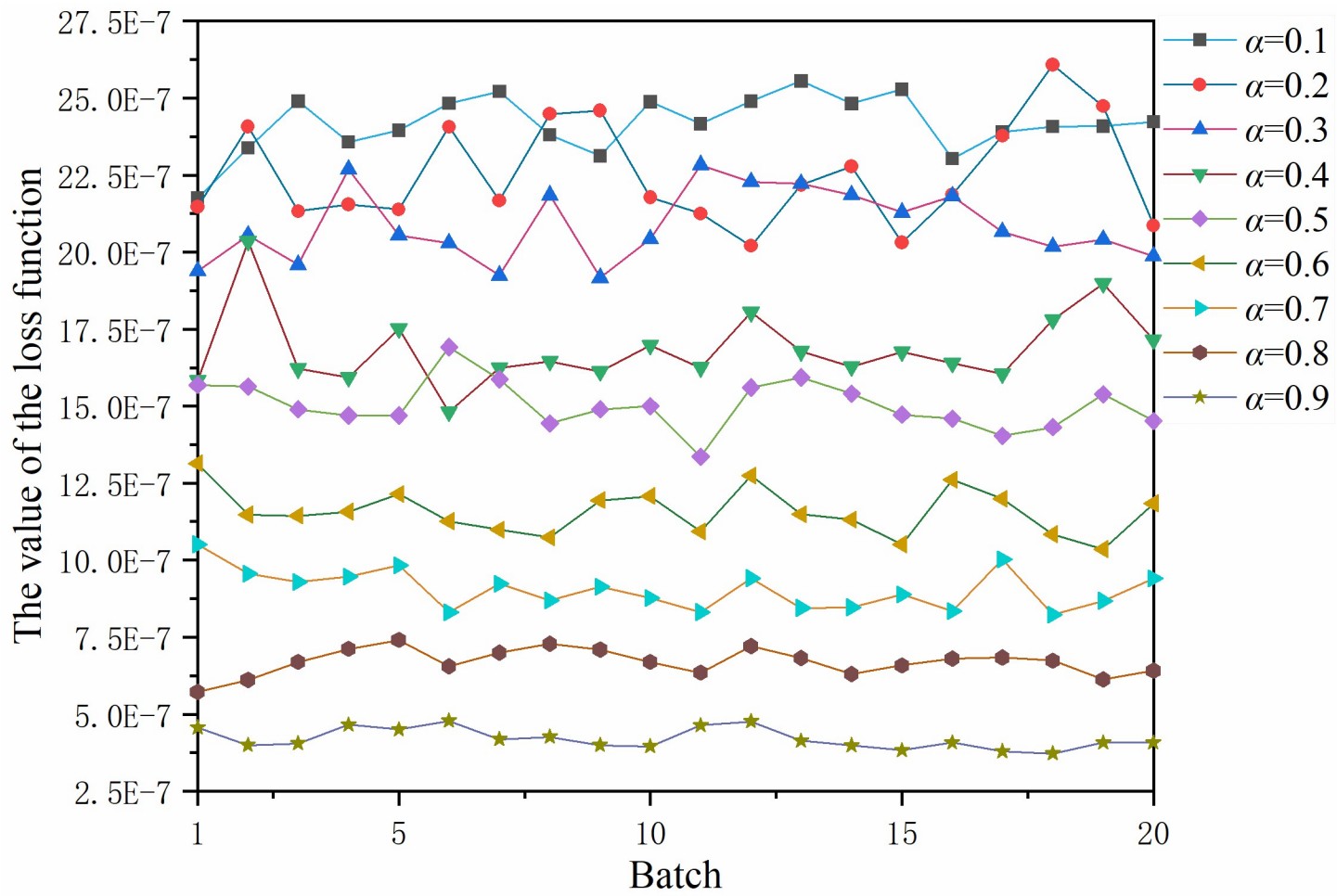

**Fig 5. The loss function value corresponding to different values of $\alpha$.**

model. To verify the generalization ability of the network model, we randomly selected 1000 sets of acoustic data from the dataset that were not involved in the model training. These data were then input into the network model with different values of $\alpha$. Predictions of the density of metamaterial unit cells were obtained. The density prediction values obtained were used to predict the sound field intensity through finite element analysis. The average error $e$ between these predicted values and the corresponding local sound field intensity values in the sample data was calculated. The validity of the metasurface inverse design method was verified. The average error e is calculated as follows:

$$e = \frac{1}{11}\sum_i^N (|P - \hat{P}|), i = 85, 86, \ldots\ldots, 95 \tag{4}$$

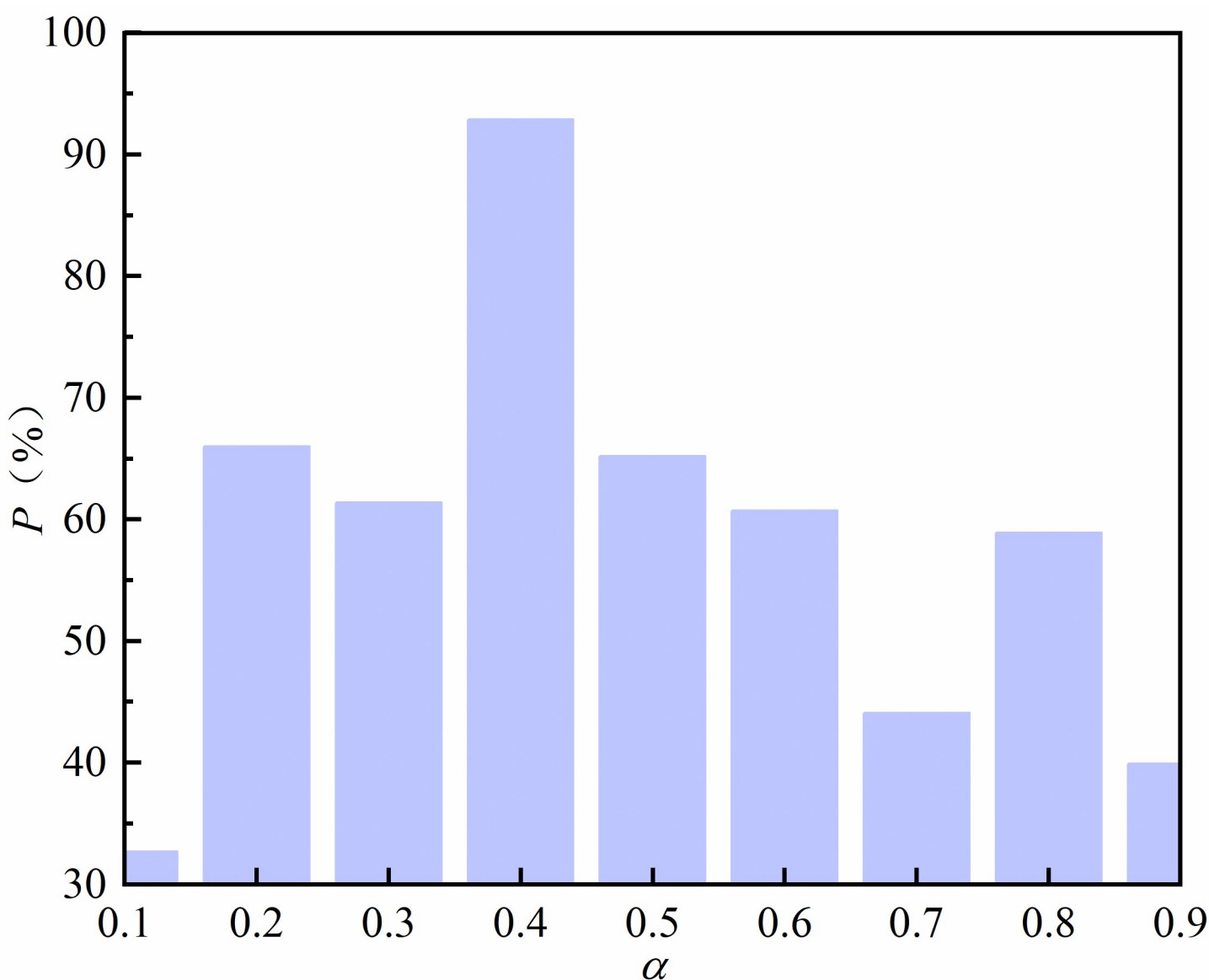

**Fig 6. Proportion of modulation intervals with mean error values less than 5 dB for different values of $\alpha$.**

In the above equation $e$ is the mean error, $P$ is the true value and $\hat{P}$ represents the predicted value. **Fig 6** illustrates that the proportion P of cases where $e$ is less than 5 dB varies with different values of $\alpha$. The optimal generalization ability was achieved at 92.9% when $\alpha = 0.4$. A comparison of **Figs 5** and **6** shows that the generalization ability of the network models corresponding to different values of $\alpha$ do not increase as the loss function decreases. **Fig 7**(A)–7(I) show the real and predicted sound field distributions at different $\alpha$ when validation parameters are the same. **Fig 7**(D) shows the sound field distribution under the optimal parameter $\alpha = 0.4$.

As shown in Fig 7(A)–7(C), when the $\alpha$ value is 0.1, 0.2, and 0.3, it represents weaker constraint capabilities. When the model is trained with more focus on local sound field characteristics. Information on the coupling relationship between multiple structural units is missing when constructing the mapping relationship between the sound field distribution to the structural parameters, which leads to inaccurate prediction of the local sound field. Fig 7(E)–7(I) illustrated the case of high constraint capacity. The network model learns more about global features. Constructing mapping relationships between sound field distributions to structural parameters focuses more on the entire sound field, which leads to neglecting local sound field feature information.

Based on the above model validation results, the model performance is best when the weighting factor $\alpha = 0.4$. **Fig 8** shows the loss function variation of the network model when $\alpha = 0.4$. Fig 9 demonstrates the curves of the true and predicted values of a certain group of acoustic field intensities at $\alpha = 0.4$. Fig 9 shows that the predicted values of the sound field intensity are in general agreement with the trend of the real values. In the local control range (i.e., within 85°~95°), the error between the predicted sound field intensity values and the true

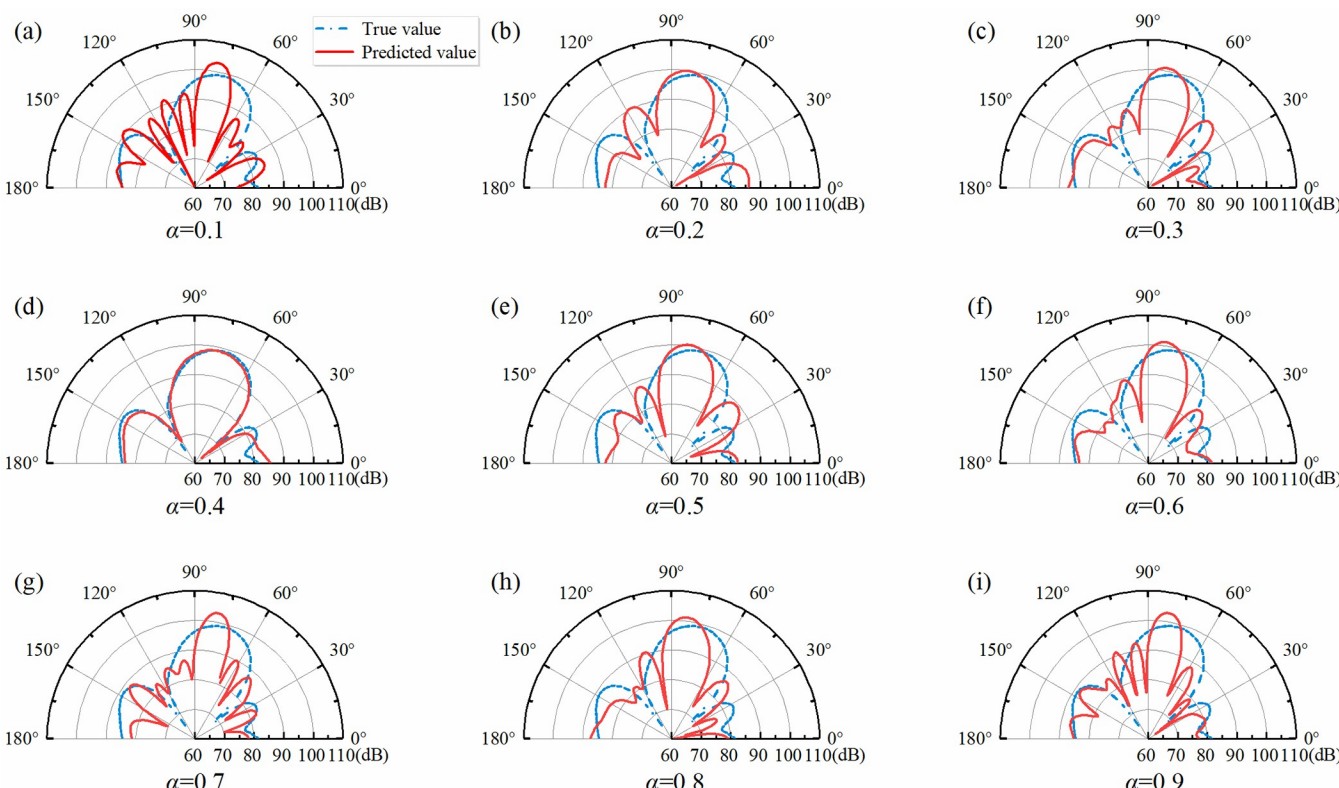

**Fig 7. (a)-(i)** respectively showed the real and predicted sound field distributions at α = 0.1~0.9 for a certain set of parameters.

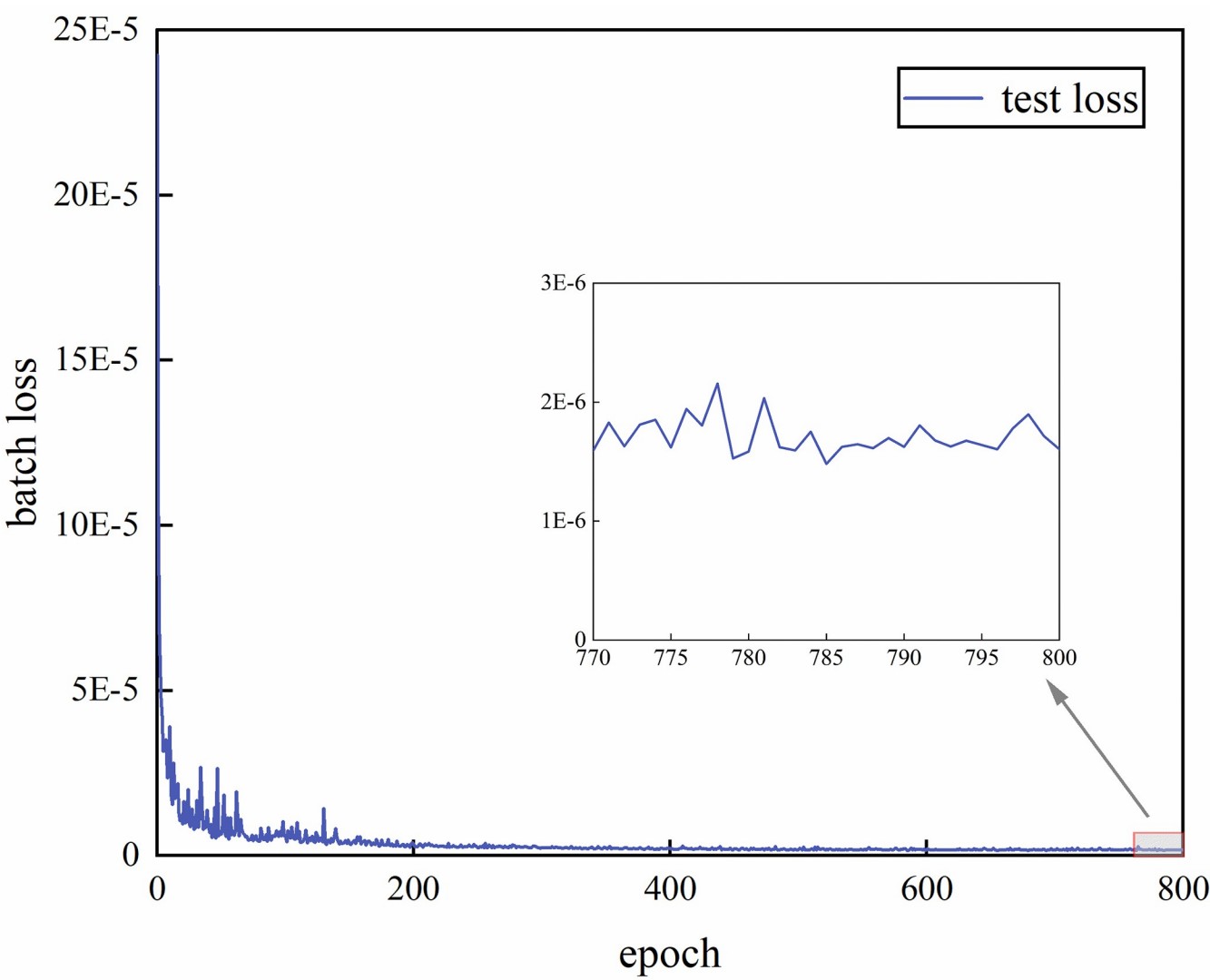

**Fig 8. Loss function curve at $\alpha$ = 0.4.**

values is approximately 1 dB. The specific distribution curve can be seen in Fig 10. As shown in Fig 11, among the randomly selected 1000 sets of data not involved in training, 92.9% of the sample data have an average error $e$ of less than 5 dB in the modulation interval. The model can establish the mapping relationship from the local acoustic field to the metasurface structural parameters, which realizes the accurate prediction of the metasurface structural parameters. The modulation of the target acoustic field intensity is realized and has a good generalization ability.

## 4.3 Loss function comparison model validation

This study introduced a comparative model validation between the K-MSE loss function and other commonly used loss functions which were SmoothL1, Quantile, Huber, MAE, and MSE. Their specific formulas were given as Eqs (5)–(9). Comparative model validation could demonstrate that the K-MSE loss function can help to construct a mapping relationship between

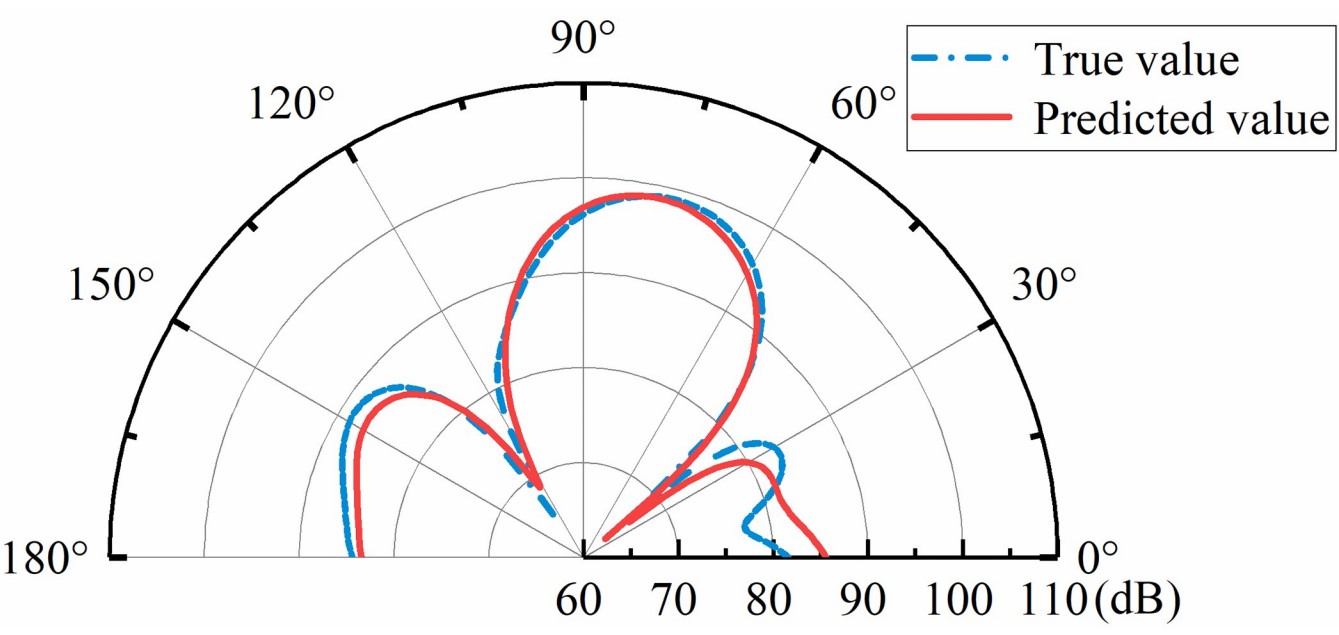

**Fig 9. Distribution curves of true and predicted sound field intensity.**

the physical structure parameters and the local sound field intensity. The training parameters of the network model were set according to **Tables 1** and **2**, while $\alpha = 0.4$. Only the loss function changed throughout the training process.

$$L_{SmoothL1} = \begin{cases} \dfrac{1}{N}\sum_{i=1}^{N} 0.5(y_i - \hat{y}_i)^2, & |y_i - \hat{y}_i| < 1 \\[2mm] \dfrac{1}{N}\sum_{i=1}^{N} |y_i - \hat{y}_i| - 0.5, & |y_i - \hat{y}_i| \geq 1 \end{cases} \tag{5}$$

$$L_{Quantile} = \begin{cases} \dfrac{1}{N}\sum_{i=1}^{N} q(y_i - \hat{y}_i), & y_i > \hat{y}_i \\[2mm] \dfrac{1}{N}\sum_{i=1}^{N} (1-q)(\hat{y}_i - y_i), & y_i \leq \hat{y}_i \end{cases} \tag{6}$$

$$L_{Huber} = \begin{cases} \dfrac{1}{N}\sum_{i=1}^{N} (y_i - \hat{y}_i)^2/2, & |y_i - \hat{y}_i| < \delta \\[2mm] \dfrac{1}{N}\sum_{i=1}^{N} \delta(|y_i - \hat{y}_i| - \delta/2), & |y_i - \hat{y}_i| \geq \delta \end{cases} \tag{7}$$

$$L_{MAE} = \frac{1}{N}\sum_{i=1}^{N} |y_i - \hat{y}_i| \tag{8}$$

$$L_{MSE} = \frac{1}{N}\sum_{i=1}^{N} (y_i - \hat{y}_i)^2 \tag{9}$$

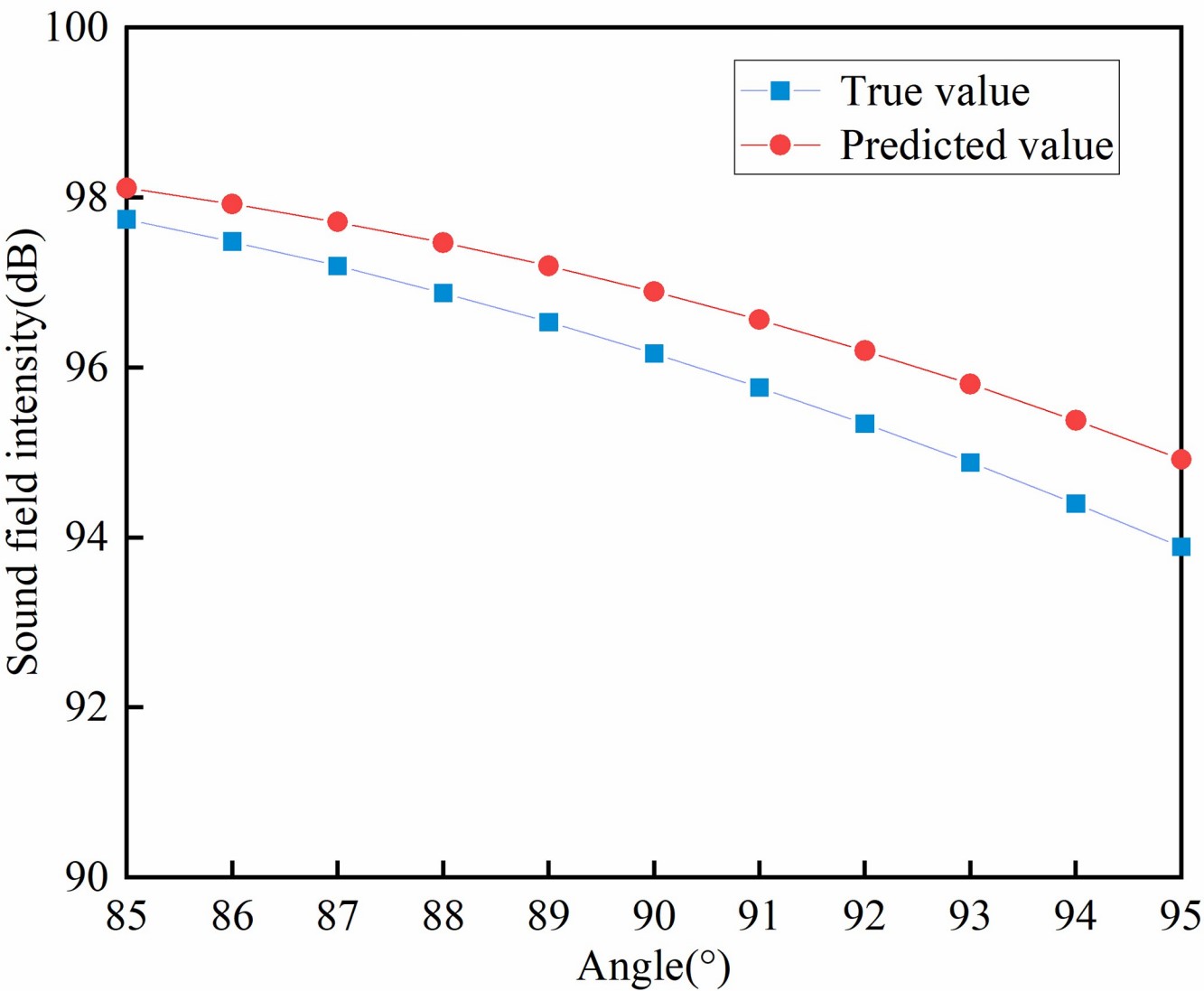

**Fig 10. Curves of real and predicted local sound field strengths.**

where $y_t$ was the true value, $\hat{y}_t$ was the model predicted value, and $N$ was the number of samples. In Eq (6), $q$ was the quantile, $q = 0.8$. In Eq (7), $\delta$ was the hyperparameter of $L_{Huber}$, $\delta = 0.5$.

The generalization ability of the network model under the six loss functions was also verified using a randomly sampled set of 1000 data sets that were not involved in training. The performance of the network model with different loss functions is shown in **Table 3**. The K-MSE loss function achieves P percentage of 92.9% when e is less than 5 dB in the target control range. It also has the smallest loss function value and the fastest convergence rate, which indicates its optimal optimization performance for the prediction network. The predicted sound field distribution for the same set of randomly selected parameters is shown in Fig 12. Fig 12 (A)-12(E) demonstrates that the errors in the main reflection angles of the predicted and real sound fields are within 15˚. The envelope trend of the sound field intensity distribution of the predicted sound field and the real sound field are similar, but the number of peaks and valleys

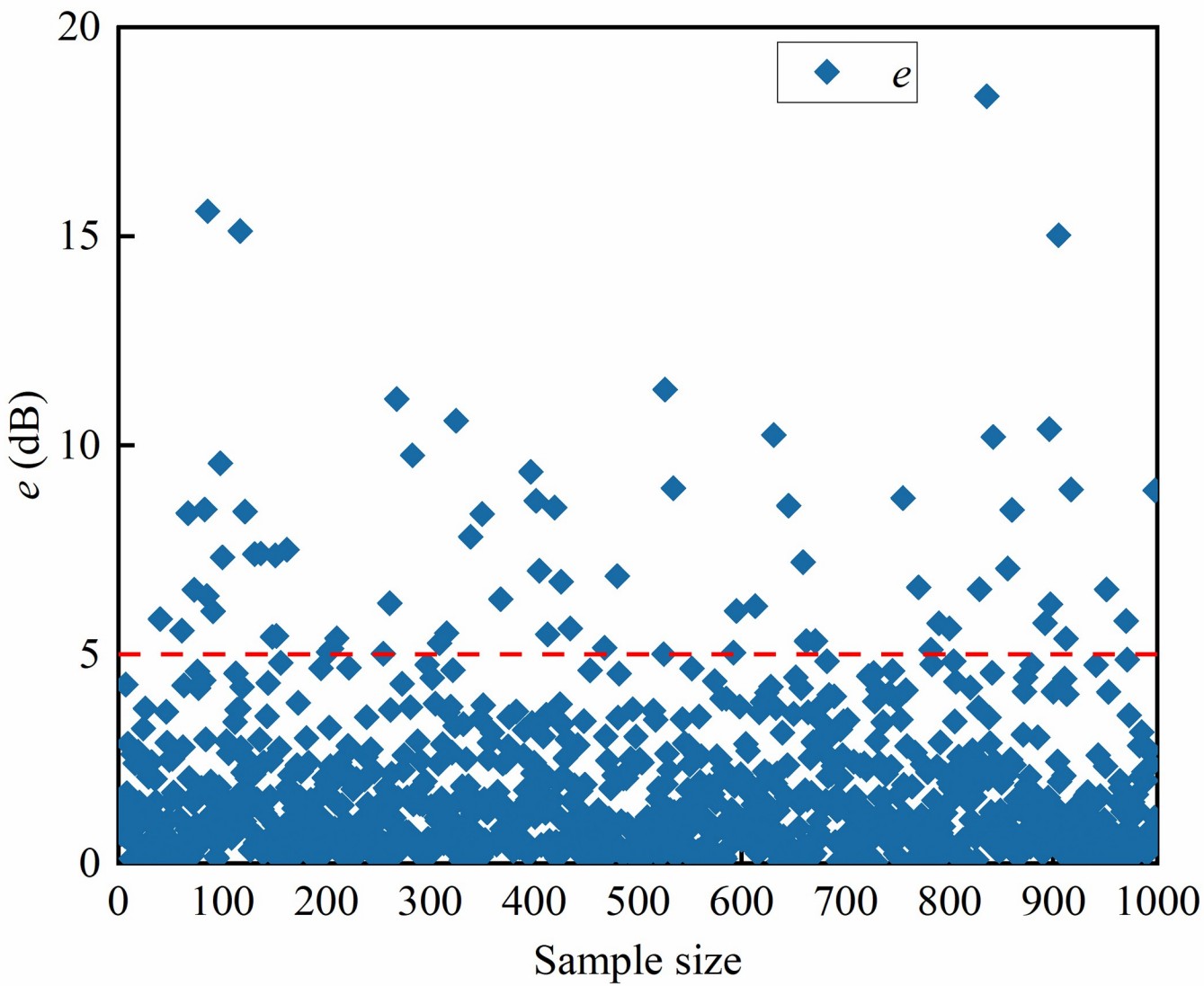

**Fig 11. The average error between the true value of the local sound field intensity and the predicted value.**

and the difference of the local sound field intensity between the two are large. As shown in Fig 12(F), the distribution of the model predicted sound field under the K-MSE loss function can have the same distribution trend as that of the real sound field, and the error of the sound field intensity value in the modulation range is about 1 dB. The K-MSE loss function helps the network model to accurately measure the nonlinearity of the error. The mapping relationship between the physical structure parameters to the sound field intensity is helped to be constructed.

**Table 3. Comparison of loss functions.**

| name | SmoothL1 | Quantile | Huber | MAE | MSE | K-MSE |
|---|---|---|---|---|---|---|
| value of the loss function | 1.7586 e-05 | 0.0017 | 1.7766 e-05 | 0.0036 | 6.1846 e-05 | 2.9612 e-06 |
| astringent batch | 622 | 697 | 531 | 592 | 492 | 475 |
| P | 28.2% | 38.2% | 59.5% | 65% | 69.1% | 92.9% |

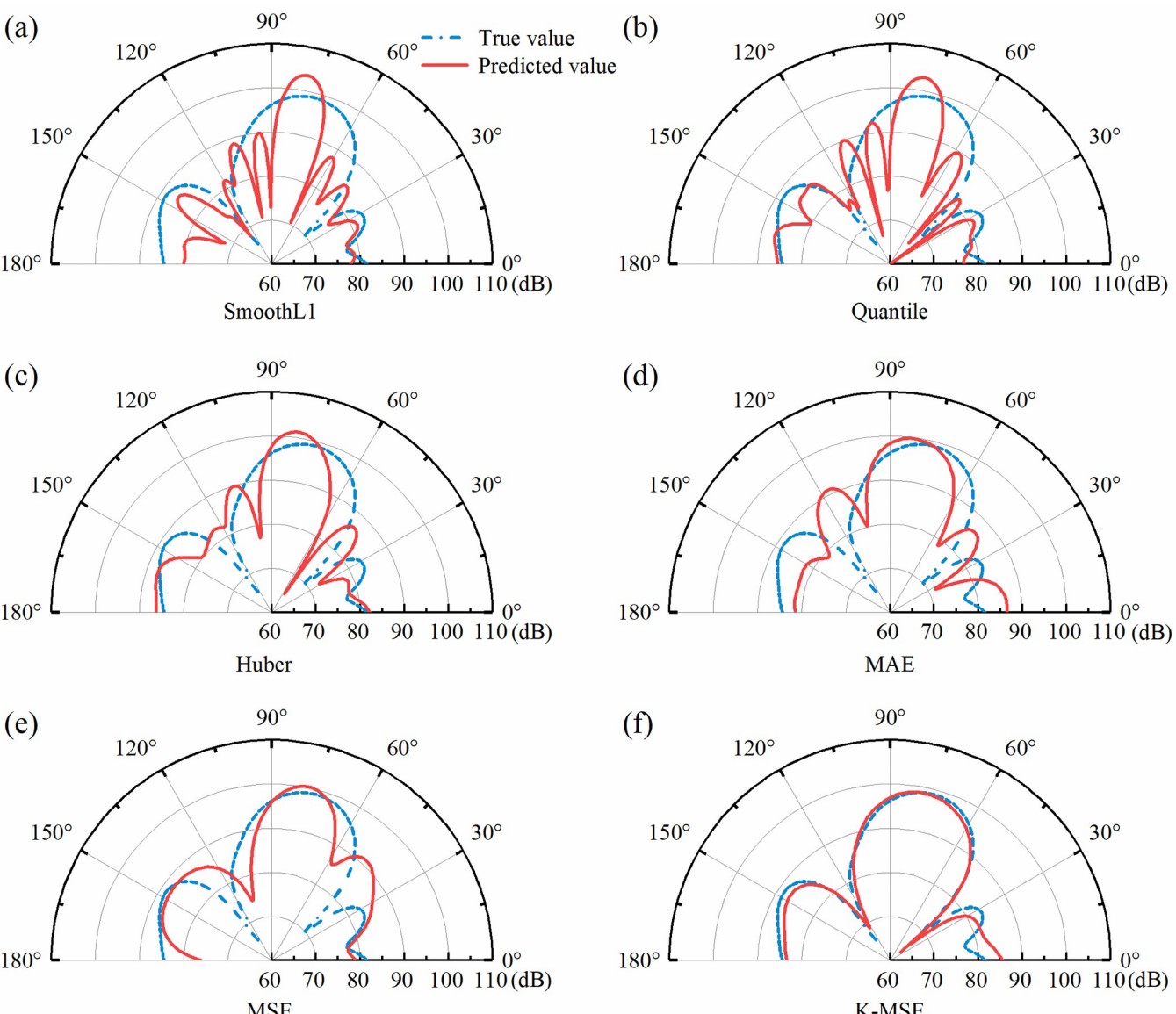

**Fig 12.** (a)-(f) refer to the actual sound field and predicted sound field distributions under different loss functions for the same parameter.

## 5. Conclusions

This paper proposed a metasurface inverse design method based on parallel deep neural networks (PDNN). The method respectively established a prediction network with local target acoustic field intensity and metasurface physical structure parameters as input and output. The global acoustic field features were used as inputs to the constraint network. The weight of the constraint network in the whole PDNN network was adjusted by adjusting the weight factor. The loss function based on the radial basis kernel function was used to train the whole network model and construct the mapping relationship from the desired sound field to the metasurface physical structure parameters. The predicted metasurface structural unit parameters could be derived from the desired acoustic field. Ultimately, the modulation of local acoustic field intensity was realized. The model validation results demonstrate that the predicted

sound field intensity curve obtained by this method closely follows the variation trend of the simulated sound field intensity curve. Additionally, it achieves a percentage of 92.9% for the sample data in which the average error between the predicted sound field intensity values and the true values falls within the specified control target range of less than 5 dB.

## Author Contributions

**Conceptualization:** Huanlong Zhao, Qiang Lv, Zhen Huang.

**Formal analysis:** Huanlong Zhao, Qiang Lv, Zhen Huang.

**Methodology:** Huanlong Zhao, Zhen Huang, Wei Chen, Guoqiang Hao.

**Supervision:** Qiang Lv, Zhen Huang.

**Writing – original draft:** Huanlong Zhao.

**Writing – review & editing:** Huanlong Zhao, Qiang Lv, Zhen Huang.

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
