## [Decision Letter · Decision Letter 0]

29 Jan 2024

PONE-D-23-30565Research on Local Sound Field Intensity Control Technique in Metasurface Based on Deep Neural NetworksPLOS ONE

Dear Dr. Lv,

Thank you for submitting your manuscript to PLOS ONE. After careful consideration, we feel that it has merit but does not fully meet PLOS ONE’s publication criteria as it currently stands. Therefore, we invite you to submit a revised version of the manuscript that addresses the points raised by the referees. Please note detailed comments from reviewer 2 is in an attached document.

We look forward to receiving your revised manuscript.

Kind regards,

Yuan Zhang, PhD

Academic Editor

PLOS ONE

Journal Requirements:

4. Thank you for stating the following financial disclosure: "The National Natural Science Foundation of China (Grant No. 61873101), the PetroChina Innovation Foundation (Grant No. 2020 D-5007-0305), and the Marine Defense Technology Innovation Center Innovation Fund (Grant No. JJ-2020-719-03-02) supported this study."

Reviewers' comments:

Reviewer's Responses to Questions

**Comments to the Author**

1. Is the manuscript technically sound, and do the data support the conclusions?

Reviewer #1: Yes

Reviewer #2: Yes

2. Has the statistical analysis been performed appropriately and rigorously? 

Reviewer #1: Yes

Reviewer #2: Yes

3. Have the authors made all data underlying the findings in their manuscript fully available?

Reviewer #1: No

Reviewer #2: Yes

4. Is the manuscript presented in an intelligible fashion and written in standard English?

Reviewer #1: Yes

Reviewer #2: Yes

5. Review Comments to the Author

Reviewer #1: This manuscript investigated the inverse design of acoustic metasurfaces based on a parallel deep neural network, where acoustic field features and the parameters of metasurfaces are the input and output, respectively. The work and the corresponding results are interesting and meaningful, yet the referee has several questions to be addressed by the authors.

1. In this manuscript, the metasurface is based on the profile of the mass density along the metasurface (or the constant gradient of the mass density more precisely). However, parametric modeling was conducted throughout the paper. How can this theoretical metasurface be physically realized in real applications? Do the authors have any proposal for this “theoretical metasurface” since it requires an increasing mass density while keeping the sound speed as a constant instead.

2. What is the physical dimension of the metasurface structural unit and the whole metasurface used in this manuscript? What is the thickness of the structural unit?

3. The authors should provide the necessary details of how the input acoustic fields are obtained. For example, what kind of physical model is used to get the acoustic field, what is the sound frequency, and etc. More importantly, what is the propagation distance for the sound intensity field since the absolute intensity level of the fields is used, but clearly the sound intensity level will be different at different propagation distance.

4. In Fig. 5, are the true values of the error for different alpha comparable? The error for the same alpha represents the degree of convergence of the model, but what is the physical meaning for the comparison between different alpha? The optimal choice of alpha=0.4 does not result in the minimum error.

5. How was the average error computed?

6. The language could be improved, and the symbols and punctuation mark should be kept consistent, for example C0 or c0 in Eq. (1), comma or semicolon in Lines 161 and 165, and etc.

Reviewer #2: This paper introduces a novel metasurface inverse design method leveraging parallel deep neural networks (PDNN). The method respectively extracts the key in formation of the acoustic field and the metasurface as the input and output of the PDNN network. With the help of the kernel loss function and the constraint performance provided by the constraint network, it establishes an inverse mapping relationship between the target acoustic field and the physical structure parameters of the metasurface. The method has certain design potential and may find applications in the field of underwater acoustic stealth. However, before accepting the publication of this paper, it is necessary to address the following questions.

6. PLOS authors have the option to publish the peer review history of their article (what does this mean?). If published, this will include your full peer review and any attached files.

Reviewer #1: No

Reviewer #2: No

---

## [Author Response · Author response to Decision Letter 0]

7 Mar 2024

Dear edit and dear reviewers,

Re: Manuscript ID: PONE-D-23-30565 and Title: Research on Local Sound Field Intensity Control Technique in Metasurface Based on Deep Neural Networks

Thank you for your letter and for the reviewers' comments concerning our manuscript entitled “Research on Local Sound Field Intensity Control Technique in Metasurface Based on Deep Neural Networks”(ID:PONE-D-23-30565). Those comments are all valuable and very helpfiul for revising and improving our paper, as well as the important guiding significance to our researches, We have studied comments carefully and have made correction which we hope meet with approval. Revised portion are marked in red in the paper. The main corrections in the paper and the responds to the reviewer's comments are as flowing:

Responds to the reviewers’s comments:

Reviewer #1: This manuscript investigated the inverse design of acoustic metasurfaces based on a parallel deep neural network, where acoustic field features and the parameters of metasurfaces are the input and output, respectively. The work and the corresponding results are interesting and meaningful, yet the referee has several questions to be addressed by the authors.

1. In this manuscript, the metasurface is based on the profile of the mass density along the metasurface (or the constant gradient of the mass density more precisely). However, parametric modeling was conducted throughout the paper. How can this theoretical metasurface be physically realized in real applications? Do the authors have any proposal for this “theoretical metasurface” since it requires an increasing mass density while keeping the sound speed as a constant instead.

Response 1: We really appreciate your professional comments on our articles. In this paper, due to the practical constraints, no actual physical implementation has been made, but the existing literature has used five-mode metasurfaces for practical applications in stealth and focusing, and for theoretical metasurfaces, it is impossible to guarantee that the sound velocity is constant, but we can use five-mode metasurfaces to avoid the influence of sound velocity.

In the theory of five-mode metasurfaces, when all sound waves are projected into the interior of the metasurface, the impedance matching of the metasurface and the medium needs to be satisfied when the perpendicular incident occurs

(1)

In Eq. (1), z is the intrinsic impedance of the metasurface, z0 is the intrinsic impedance of the medium, ρ and ρ0 are the densities of the metasurface and the incident medium, respectively, c and c0 represent the velocity of sound of the metasurface and the incident medium:

(2)

Eq. (2) illustrates that while controlling the speed of sound, the density needs to change in the opposite direction to achieve impedance matching. The density and speed of sound of natural materials generally change with the same trend, and in the case of a change in the speed of sound, there is no guarantee that the impedance will not change, and the speed of sound cannot be guaranteed to be constant. Therefore, to meet the conditions for the existence of full transmission, only artificial materials can achieve it. The ideal density distribution of the reflective five-mode metasurface is

(3)

In Eq. (3), θr is the reflection angle, L is the metasurface length, and H is the normal thickness of the metasurface.

2. What is the physical dimension of the metasurface structural unit and the whole metasurface used in this manuscript? What is the thickness of the structural unit?

Response 2: We appreciate your professional comments on our article. The focus of this article is on the prediction of the physical structure parameters of the metasurface, and there is no restriction on the internal structure of the metasurface structural units; the normal thickness of the metasurface is 0.12 m, the length is 1 m, and the length of each metasurface structural unit is 0.04 m. We have added the specific settings into the article at line 82,97 and 98. We thank you very much for your careful reading.

3. The authors should provide the necessary details of how the input acoustic fields are obtained. For example, what kind of physical model is used to get the acoustic field, what is the sound frequency, and etc. More importantly, what is the propagation distance for the sound intensity field since the absolute intensity level of the fields is used, but clearly the sound intensity level will be different at different propagation distance.

Response 3: We appreciate your professional comments on our article. The physical field of the water and the hypersurface is set as pressure acoustics, the acoustic frequency is the frequency of the incident wave, and the backing plate is set as solid mechanics; we establish the acoustic field model with the help of COMSOL simulation software, and the main work of this paper is to verify whether the deep learning modeling method can effectively realize the regulation of the local acoustic field with the help of the acoustic field model, and the results show that based on the deep learning modeling method, it can effectively realize the reverse design and thus achieve the regulation of local acoustic field of the hypersurface. The results show that based on the deep learning model method, the hypersurface inverse design can be effectively realized, and then realize the local acoustic field control. Thank you very much for your question.

4. In Fig. 5, are the true values of the error for different alpha comparable? The error for the same alpha represents the degree of convergence of the model, but what is the physical meaning for the comparison between different alpha? The optimal choice of alpha=0.4 does not result in the minimum error.

Response 4: Thank you very much for your question. The error truth values of different alpha are comparable, and only the value of alpha changes when validation with different alpha is performed. In Fig. 5, the same data set is used to validate the models built with different alpha values during validation, and the performance of the models corresponding to different alpha values can be seen to be superior or inferior based on the difference in the true value of the error; because of the coupling relationship between multiple hypersurface units, the local sound field will be affected by the whole sound field. In this paper, a parallel neural network is used, one model network extracts the global sound field feature information, and the other network extracts the local sound field feature information. In this paper, we use the alpha value can be used to adjust the constraint strength of the global sound field to the local sound field, and we introduce it into the kernel loss function and it plays a role in the model, we need to find out the optimum value of alpha to make the model performance optimal and maximize the accuracy of the prediction. We need to find the optimal alpha value to optimize the model performance and maximize the accurate prediction, we validate the role of alpha value on the model in the paper, and from the results, we can see that the appropriate alpha value has a greater impact on the accuracy of the deep learning model. Thank you for your careful reading.

5. How was the average error computed?

Response 5: Thank you for your question and I apologize for the confusion here. The formula for calculating the average error is as follows:

In the above equation e is the mean error, P is the true value and P(ˆ) represents the predicted value. We have added the formula for calculating the mean error to the article at line 205-208. Thank you for your careful reading.

6. The language could be improved, and the symbols and punctuation mark should be kept consistent, for example C0 or c0 in Eq. (1), comma or semicolon in Lines 161 and 165, and etc.

Response 6: Thank you for your question, and sorry for the trouble here. We've changed the parameters or punctuation to lines 71, 165, and 169. We are very sorry for our careless mistakes. Thanks for the reminder. We've rechecked and corrected the article to make punctuation consistent throughout the article. Thank you for your correction.

Reviewer #2: This paper introduces a novel metasurface inverse design method leveraging parallel deep neural networks (PDNN). The method respectively extracts the key in formation of the acoustic field and the metasurface as the input and output of the PDNN network. With the help of the kernel loss function and the constraint performance provided by the constraint network, it establishes an inverse mapping relationship between the target acoustic field and the physical structure parameters of the metasurface. The method has certain design potential and may find applications in the field of underwater acoustic stealth. However, before accepting the publication of this paper, it is necessary to address the following questions.

1. The state-of-the-art review in the introduction section is a bit poor. In particular, the description of metasurfaces and pentamode metamaterials is a bit too fast citing only a few papers for each mentioned field and does not give proper credit to previous literature contributions. The manipulation of scattering characteristics in the paper aligns with the concept of coding metasurfaces [Natl. Sci. Rev. 2022, 9, nwac030; Phys. Rev. Appl. 2022, 17, 034019; Small 2024, 2308349, DOI: 10.1002/smll.202308349]. The conceptual framework for the design of the pentamode metamaterial also requires richer literature support [J Acoust Soc Am 2009; 125: 839-849; J Sound Vib 2019; 443: 238–252; J Mech Phys Solids 2021; 152:104407].

Response 1: We sincerely thank you for your valuable comments. We have carefully reviewed the introductory section and added more references to metasurfaces and penta-mode metamaterials in the introduction section of the revised draft below：

1. Achromatic metasurfaces by dispersion customization for ultra-broadband acoustic beam engineering (doi: 10.1093/nsr/nwac030);

2. Broadband Coding Metasurfaces with 2-bit Manipulations (doi: https://doi.org/10.1103/PhysRevApplied.17.034019);

3. Enhanced Broadband Manipulation of Acoustic Vortex Beams Using 3-bit Coding Metasurfaces through Topological Optimization (doi: 10.1002/smll.202308349);

4. Acoustic metafluids (doi: https://doi.org/10.1121/1.3050288);

5. Convective correction of metafluid devices based on Taylor transformation (doi: https://doi.org/10.1016/j.jsv.2018.11.047);

6. Customized broadband pentamode metamaterials by topology optimization (doi: https://doi.org/10.1016/j.jmps.2021.104407)

7. Design and experimental verification of a water-like pentamode material (doi: 10.1063/1.4973924)

We have introduced the above literature in the article to complete the introduction section. Thank you for reading.

2. This paper is limited to the optimization design of equivalent parameters and does not delve into more complex pentamode metamaterial configuration designs. To enhance the completeness of the paper, it is suggested to provide potential structural instances based on the designed parameter distribution. Even referencing cases from the literature would be beneficial in demonstrating the effectiveness of the design and the rationality of the designable parameters.

Response 2: Thank you very much for this suggestion, we agree with your suggestion. The work of this paper is to use deep learning to predict the structure parameters of the metasurface, and then realize the local acoustic field control. Due to the fact that the actual design of the metasurface structure unit is long in modeling time and requires high accuracy, the current experimental conditions are difficult to meet, and the specific metasurface structure is not involved, in the follow-up work, we will apply the deep neural network to the study of different metasurface structures and a wider range of sound field control, and hope to truly realize the sound field control in the future. Thank you very much for your suggestions.

3. Since this paper does not involve the design of actual configurations and lacks experimental validation, the use of the term "experiment" in the main text is inappropriate.

Response 3: Thank you very much for your suggestion. Since there is no actual configuration involved, there is indeed an inappropriate problem with the use of "experimental" in the body, and we have modified the word "experimental" to "model validation" in the body of the article at 59,146,224,238,239,242 and 280. Thank you very much for pointing out the problem.

4. Does the scattering wave control strategy based on machine learning offer the advantage of faster and more convenient implementation? If so, does this attribute hold potential for applications in three-dimensional metasurface design and even curved metasurface design? Please provide further insights on this matter.

Response 4: Thank you for your questions. The scattered wave control strategy based on machine learning has the advantage of faster and more convenient implementation, when the machine learning model establishes the mapping relationship between input and output, the physical structure parameters of the metasurface can be obtained in a few seconds, which is much faster than the traditional way of establishing a simulation model. Machine learning has the potential to be applied in 3D metasurface design and even surface design, and it is a method of automatically learning the connection between the input data and the target data from examples from past experience. On the premise of having relevant datasets, the machine learning model can establish a high-dimensional mapping relationship between 3D/surface metasurface parameters and the sound field, and can quickly obtain the desired structural parameters or target sound field, which is helpful for high-dimensional, black-box, and time-consuming optimization problems, and reduces the time and labor cost of repeated experiments. Thank you very much for your questions.

5. This manuscript primarily focuses on the equivalent parameter design of underwater low-detectability metasurfaces. It is recommended to enhance its functionality, for instance, by comparing the reflection energy of this scattering metasurface with that of a fully reflective metasurface to evaluate the metasurface's acoustic target characteristics.

Response 5: We very much agree with you, and your suggestion provides a direction for our next research, which will further focus on the performance of metasurfaces in terms of energy. Detection and other fields play an important role, of course, not only in terms of energy, we also consider combining specific metasurface geometries to verify the performance of local control in actual scenarios, and also expect to achieve a wider range of sound field control, in the future work, we will enhance its function from including but not limited to these aspects. Thank you very much for your suggestion, which provides us with a good idea.

---

## [Decision Letter · Decision Letter 1]

12 Mar 2024

Research on Local Sound Field Intensity Control Technique in Metasurface Based on Deep Neural Networks

PONE-D-23-30565R1

Dear Dr. Lv,

We’re pleased to inform you that your manuscript has been judged scientifically suitable for publication and will be formally accepted for publication once it meets all outstanding technical requirements.

Kind regards,

Yuan Zhang, PhD

Academic Editor

PLOS ONE

Additional Editor Comments (optional):

Reviewers' comments:

Reviewer's Responses to Questions

**Comments to the Author**

1. If the authors have adequately addressed your comments raised in a previous round of review and you feel that this manuscript is now acceptable for publication, you may indicate that here to bypass the “Comments to the Author” section, enter your conflict of interest statement in the “Confidential to Editor” section, and submit your "Accept" recommendation.

Reviewer #1: All comments have been addressed

Reviewer #2: All comments have been addressed

2. Is the manuscript technically sound, and do the data support the conclusions?

Reviewer #1: Yes

Reviewer #2: Yes

3. Has the statistical analysis been performed appropriately and rigorously? 

Reviewer #1: Yes

Reviewer #2: Yes

4. Have the authors made all data underlying the findings in their manuscript fully available?

Reviewer #1: Yes

Reviewer #2: Yes

5. Is the manuscript presented in an intelligible fashion and written in standard English?

Reviewer #1: Yes

Reviewer #2: Yes

6. Review Comments to the Author

Reviewer #1: (No Response)

Reviewer #2: After a rigorous review, the authors of the paper have comprehensively addressed the concerns raised by the reviewer, and the overall writing of the paper is rigorous. Therefore, the reviewer has no further questions.

7. PLOS authors have the option to publish the peer review history of their article (what does this mean?). If published, this will include your full peer review and any attached files.

Reviewer #1: No

Reviewer #2: No

---

## [Editor Report · Acceptance letter]

18 Mar 2024

PONE-D-23-30565R1 

PLOS ONE

Dear Dr. Lv, 

I'm pleased to inform you that your manuscript has been deemed suitable for publication in PLOS ONE. Congratulations! Your manuscript is now being handed over to our production team.

Kind regards, 

on behalf of

Professor Yuan Zhang 

Academic Editor

PLOS ONE